# Reconfigurable Metasurface: Enabling Tunable Reflection in 6G Wireless Communications

**DOI:** 10.3390/s23229166

**Published:** 2023-11-14

**Authors:** Monisha Selvaraj, Ramya Vijay, Rajesh Anbazhagan, Amirtharajan Rengarajan

**Affiliations:** School of EEE, SASTRA University, Thanjavur 613401, India; monisha@sastra.ac.in (M.S.); rajesha@ece.sastra.ac.in (R.A.); amir@ece.sastra.edu (A.R.)

**Keywords:** metasurface, 6G communication, reflecting intelligent surface, phase reconfigurable metasurface, reflecting surface

## Abstract

With the continuous advancement of technology, there is an increasing need for innovative solutions that can handle complex applications such as haptic communications, Internet of Things for smart cities, automation, and manufacturing. One technology that has received much attention is the phase reconfigurable metasurface for reconfigurable intelligent surfaces (RISs). The RIS demands low-power consumption, simple configuration, angular stability, and polarization insensitivity. The use of phase reconfigurable metasurfaces provides benefits such as low cost, low power consumption, and improved communication coverage and quality. This article introduces a reconfigurable combined-loop metasurface that can effectively manipulate phase reflection. This is achieved by incorporating four PIN diodes between two meta-atoms of a 2 × 2 periodic array within a single-layer metallic structure. By controlling the state of the PIN diodes, which can be switched into 16 different states, the metasurface can achieve various phase reflections. The proposed structure has validated a 32× 32 metasurface through numerical simulations and experiments that exhibit promising results, demonstrating its potential for use in 6G applications.

## 1. Introduction

Over the past few years, there has been a significant increase in the demand for high-speed and reliable wireless communication. This demand has been driven by the growth of Internet of Things networks and technological advancements such as 5G and beyond [1]. Researchers and engineers are working hard to develop new technologies that can enhance the capacity and coverage of wireless networks in response to this growing demand. One of the most promising emerging technologies in wireless communications research is the reflecting intelligent surface (RIS). Also known as a software-controlled metasurface, the RIS has been identified as a solution for achieving intelligence in wireless channel and propagation environments in 5G and 6G communication systems [2].

The RIS is a passive array of electromagnetic elements that can reconfigure wireless propagation by intelligently reflecting signals and controlling the reflected waves’ phase, amplitude, and polarization. These surfaces can effectively manipulate the wireless propagation environment, enhancing signal quality and coverage. The RIS is created by integrating intelligent materials into conventional objects such as walls or windows, allowing them to control and manipulate transmitted signals. An RIS is a two-dimensional surface with many sub-wavelength elements that can be electronically controlled to reflect, refract, or manipulate [3,4]. The basic architecture of an RIS is shown in Figure 1.

There are several adequate works on RIS-related secure communication [5,6,7], reducing power consumption [8] using optimization methods and simulations. Most studies in literature concentrate on the theoretical side of an RIS and its simulation. However, there has yet to be much progress made in RIS design or hardware fabrication up until now. The literature section of this paper will comprehensively explain both transmissive and reflective forms of an RIS to realize the constraints of RIS hardware design.

In early RIS designs, the authors developed generalized sheet transition requirements for the average electromagnetic fields throughout a meta-film [9]. They computed the meta-films’ transmission and reflection coefficients. A “smart” and/or “controllable” surface was achieved by manipulating the scatters’ polarization densities within the meta film. Utilizing spherical magneto-dielectric particles, a controlled surface was obtained. The tunability of spherical magneto-dielectric particles may be limited, especially in specific frequency ranges, which can restrict their effectiveness in dynamically controlling surface waves and electromagnetic fields over a broad spectrum. The size of spherical magneto-dielectric particles may impose constraints on their applicability in specific applications, especially those requiring miniaturization and integration into compact devices or systems.

A “Digital metamaterial” that uses “coding metamaterial” was propped in [10], which is made up of two-unit cells with a 0 and π phase response. Phases 0 and π can be arranged sequentially to produce different functionalities. The authors eventually came up with “programmable metamaterial’. Implementing coding schemes within metamaterial structures may pose challenges in practical implementation and integration with existing communication protocols and standards, potentially limiting their seamless incorporation into commercial communication networks. So, a technique called a hybridized resonator was introduced, and the authors of [11] suggested a binary state controllable phase reflector. This technique is robust and produces fluctuations caused by an adjustable mechanism. The control mechanism can be implemented by adding electronic components to the surface. Varactor diodes are a unique type of diode that can change capacitance in response to voltage changes. Adjusting the unit cell’s capacitance can modify the impedance and, as a result, control the reflected waves.

This proposed work focuses on the structure that supports multi-features to develop a reflecting intelligent surface physical layer design. The prime contribution is the choice of a structure and its integration to achieve the following:✓ Single-layer phase reconfigurable reflecting metasurface structure to address the design complexity;✓ Good angular stability and polarization sensitivity to address the minimum coverage issue from existing RIS systems;✓ Phase reconfiguration through an active device maintains the ideal phase difference between the unit cells.

### Literature Survey

The authors of [12] modified the electromagnetic characteristics of a metal ground plane by applying a periodic surface texture. It was possible to create a tunable impedance surface where the bias voltage regulates the reflection phase and the resonant frequency by including varactor diodes into the texture. It was possible to adjust the surface to produce a phase gradient that could guide a reflected beam by +/− 40°. The authors proposed a varactor diode-based RIS in the sub-6 GHz frequency range [13]. The suggested design operated at a frequency of 3.5 GHz. The cells that made up the construction had 2430 units. The reflected beam was rearranged with the use of a varactor diode. Another varactor diode-based RIS was suggested in [14] and contains 1100 controlled elements operating at 5.8 GHz. They proposed a practical approach for customizing the RIS over the air. A power boost of 27 dB was noted in the short-distance measuring scenario. The authors assessed the RIS’s performance over a 500 m outdoor long-distance course. The RIS performed well in the outside measuring setting as well. A varactor diode with a split ring was considered in [15]. With the suggested unit cell, a phase shift 277° at 24.5 GHz and a minimum reflection amplitude of 0.5 were attained. A varactor diode-based RIS was proposed in [16]. This model was tailored for the RIS-assisted Ambient Backscatter Communication system, and the suggested RIS considerably increased the system performance. Even though the performance varactor-based RIS is better, it consumes a high degree of power during the control and modulation processes, contributing to less energy efficiency and low battery life for wireless devices and systems.

The other controlling device, a PIN diode, was considered to design RIS. PIN diodes offer the advantage of low loss, high durability, compact size, improved energy efficiency with longer battery life, and low power consumption. PIN diodes were utilized to build an electrically reconfigurable reflect array, as per the authors’ proposal in [17], which involved massive electronically reconfigurable microstrip patches. Then, in [18], a 2-bit low-cost, high-gain RIS was suggested. The suggested structure gained 21.7 dBi at 2.3 GHz and 19.1 dBi at 28.5 GHz. The authors provided the phase and magnitude response of the RIS for four-element combinations. The authors of [19] described an RIS based on mm-Wave PIN diodes. After carefully examining the PIN-based metasurface, the authors acquired a broad bandwidth near the 28 GHz frequency range. After that, the authors created the unit cell and confirmed its features.

In [20], the authors proposed a PIN diode-based RIS working at 5.8 GHz. Computer vision was utilized to aid the RIS-based beam tracking. A camera was used to obtain visual information about the surrounding environment of the RIS, and the data captured by the camera was used to reflect the beam in the desired direction. The suggested antenna was tested in close field and far file settings. In [21], a novel kind of 2-bit unit cell was put forward. Realizing an RIS with this design was relatively easy. In [22], a brand-new RIS type operating at 5.8 GHz was proposed. A 16 × 10 element array was considered to obtain the RIS after a basic patch, and a PIN diode was added to the unit cell. The unit cell had a parasitic patch on its 1-bit working mechanism. The authors numerically analyzed the performance of the unit cell. The magnitude of the reflection was negligible between the two states, and the phase difference between the two states was 180°. A 64-element, 2-bit high-accuracy RIS was proposed in [23], where FPGA could control each element individually. Each unit cell had two PIN diodes, and an FPGA could control each PIN diode. The proposed RIS could be deflected up to 30°. In [24], a 1-bit RIS was proposed to aid wireless communications to overcome path loss and shadowing. In [25], the authors proposed an improved path-loss model suitable for RIS-aided wireless communications.

According to the literature survey, overcoming issues like high power consumption, complex configuration, angular instability, and polarization insensitivity is vital. The significance of RIS design is as follows:✓ For practical activities, that the RIS should use the least amount of power is essential;✓ To reduce design complexity, the RIS should be configured;✓ RIS can take all incoming waves regardless of their polarizations, and the RIS should handle angles of incidence. The maximum suggested RISs are sensitive to the angle of incidence and are not polarization independent.

Hence, designing RIS gives an enormous scope for this research. Maintaining the ideal phase difference between the unit cells is vital for RIS design. On the other hand, minute manufacturing flaws can drastically alter the phase difference. Therefore, multifunctional features of RIS are expected for wireless communications.

## 2. Design Flow of the Proposed Structure

### 2.1. Fundamental Architecture of Proposed Unit Cell

The Metasurface structure is divided into three layers: top, middle, and bottom (Figure 2). Each layer affects the frequency response differently.

The top layer of the electromagnetic device consists of conducting elements arranged in a specific manner to achieve the desired frequency response. The proposed design structures employ square loop metasurface because of their high level of customize ability to adjust the geometry of the loops to meet our specific design requirements. Further, it is integrated with an octagonal loop metasurface to reduce coupling between neighboring elements and significantly enhance the metasurface’s overall performance. This integrated loop structure ensured that the electromagnetic waves reflected from the source were polarized in a specific direction, considerably reducing signal degradation, and improving overall quality. Also, using a tripole metasurface with diodes in the integrated loop helps to adjust the phases of the waves, precisely steering them towards the intended receiver, even in the presence of obstacles or interference.

In this structure, copper (Cu) is the preferred metal layer due to its high conductivity and low cost. The middle layer acts as a substrate that controls the propagation of the electromagnetic (EM) wave. It separates the top and bottom layers, and the substrate selection is crucial for the metasurface’s proper functioning. The relative permittivity (ε_r_) and thickness (t) are the parameters that affect the frequency response of the substrates. For high-frequency applications, materials with lower relative permittivity are preferred. Rogers RT/Duroid 5880, with an ε_r_ value of 2.2, is the substrate selected for this device. The bottom layer acts as a conductive plane at the back of the electromagnetic device. However, no conductive plane is assigned during the radar cross-section simulation of the unit cell. In the complete architecture design of RIS, a conductive plane is placed at the back to block the transmitted wave and provide effective total reflection. Figure 3 shows the structure of the meta-atom, and Table 1 shows the meta-atom parameters and values.

### 2.2. Reconfigurable Metasurface

Researchers have explored using metasurfaces and metamaterials to achieve phase modulation and reconfigurability in phase-reconfigurable systems. In a study conducted by [26], a phase reconfigurable metasurface was designed using PIN diodes to dynamically control the phase of electromagnetic waves. The literature survey also reveals that PIN diodes are widely used in phase reconfigurable metasurface. Placing diodes within the metasurface is crucial to achieve the desired functionality and manipulating the current distribution. The desired current distribution can be determined by the functionality that needs to be achieved. Positioning the diodes strategically along the metasurface allows for the current flow to be manipulated, resulting in pattern reconfigurability. It allows for the control of the beam, depending on the ON/OFF state of each pin diode. Placing pin diodes in a metasurface allows for the equivalent length of slots to be adjusted, altering the current distribution within the structure. This can be achieved by strategically placing the pin diodes in specific locations along the metasurface, rearranging the current distribution and enabling the desired pattern reconfigurability due to the current distribution of a single element (Figure 4), trying to place the PIN diode on the right edge of the unit cell. That leads to an electrical length change and gives frequency reconfiguration. So, to maintain the electrical length and alter the current flow, instead of using a PIN diode in the meta-atom, it is connected with the metasurface using PIN diodes with the reference of the current distribution.

The current distribution of the proposed work shown in Figure 4 helps to adopt the diodes between the two elements. The placement of the diode is based on the high current flow in the metasurface. Therefore, the D1 is placed between the top edges of the first- and second-unit cells due to the high current flow at the top edge of the first-unit cell. Likewise, all the diodes are placed according to the symmetric current distribution of the proposed work shown in Figure 5. Four diodes (D1, D2, D3, D4) are used at the high current distribution of the 2 × 2 metasurface. The diodes between the two elements of the metasurface are shown in Figure 5.

### 2.3. Switch Controller

The ability to control the switching condition of a PIN diode using an Arduino has been made possible with a microcontroller-controlled embedded biasing network. As detailed in [27], this circuit design eliminates the need for a DC power supply and various lumped components, reducing the setup’s overall cost, weight, and complexity. The microcontroller, such as the ATMEGA 2560, controls the PIN diode switching [28]. It generates the necessary control voltage to bias the PIN diode, which controls its RF switching state. The essential advantage of this approach is that the microcontroller can toggle the control voltage between “ON” and “OFF” states with precision and programmability. This feature provides flexibility in controlling the timing of the PIN diode switching, allowing for customization based on specific application requirements.

Furthermore, an Arduino board offers a user-friendly and accessible platform for programming and controlling the PIN diode. Additionally, the Arduino programming language simplifies the implementation of the control logic. Overall, the use of an Arduino and a microcontroller-controlled embedded biasing network provides an effective and efficient method for controlling the switching condition of PIN diodes. In this work, the PIN diode is governed by the controller, and their states can be controlled by a keypad through a level shifter to control the forward voltage of the PIN diode, as depicted in the block diagram in Figure 6.

## 3. Performance Simulation

The 7 GHz to 24 GHz frequency range can support wide-area coverage and high capacity, especially with the new technological advances of 6G. For supporting wide-area use cases, X-band (8–12 GHz) and Ku-band (12–18 GHz) are globally discussed [29,30]. Hence, this proposed work’s frequency covers the Ku-band with an operating center frequency of 15.4 GHz.

The architecture of an RIS consists of multiple layers, each with a specific purpose of enhancing communication performance. The first layer of the RIS architecture comprises passive reflecting elements, while the second layer comprises copper plates placed below the passive reflecting elements. The third and final layer is the intelligent control module, which is responsible for controlling and optimizing the performance of the entire RIS system.

### 3.1. First Layer

The first layer contains an array of elements with low cost and operates as reflecting surfaces for radio signals. They are designed to efficiently reflect and redirect signals in the desired directions, enhancing signal strength and coverage for wireless environments.

#### 3.1.1. Investigation of Unit Cell

Metasurfaces possess several key characteristics that make them highly suitable for manipulating electromagnetic waves. The emergence of metasurface technology has opened up new opportunities in various applications by allowing for the precise manipulation of electromagnetic waves in multiple dimensions. Furthermore, metasurfaces can be designed to exhibit unique phenomena that are not typically observed in conventional materials. For instance, metasurfaces can enable negative-angle refraction, where light is bent in a direction opposite to that predicted by Snell’s law. Due to their unique metasurface characteristics, an analysis of the effective medium theory is essential. However, in (1–3), the retrieval process for the effective medium parameters [31] is presented.
(1)$$\mu_{r}\approx \frac{\lambda (1-S_{21}+S_{11})}{j\pi d(1+S_{21}-S_{11})}$$
(2)$$\varepsilon_{r}\approx \mu_{r}+j\frac{\lambda S_{11}}{\pi d}$$
(3)$$\eta_{r}\approx \sqrt{\frac{\lambda }{j\pi d}\left\{\frac{{\left(S_{21}-1\right)}^{2}-{S_{11}}^{2}}{{\left(S_{21}+1\right)}^{2}-{S_{11}}^{2}}\right\}}$$

Figure 7a–c illustrate the effective permeability (μ_r_), permittivity (ϵ_r_), and refractive index ($\eta_{r}$). In the effective medium, negative peaks occur at an operating frequency (15.4 GHz) with substrate t and wavelength λ thickness. Figure 7a–c display that the permittivity, permeability, and refractive index curve deliver negative real and imaginary values at 15.4 GHz. So, the design is a metasurface and confirms that the proposed system satisfies the left-handed behavior.

Metasurface angular stability analysis studies how metasurfaces function when subjected to different angles of incidence or viewing. This analysis is crucial in determining the effectiveness of metasurfaces in manipulating electromagnetic waves. It ensures the consistent and reliable performance of metasurfaces regardless of the angle at which they are viewed or interacted. The proper selection of the structure helps to reduce variations in resonance frequency. The angle mean deviation (AMD) [32] can be used to calculate the frequency deviation, and it can determine the angular stability for TE and TM modes of operation under varying incidence angles using (4).
(4)$$\delta f_{as}=\sum_{i=0}^{n}\frac{f_{r}^{normal}-f_{r}^{a_{i}}}{nf_{r}^{normal}}\times 100\%$$
where $f_{as}$, $f_{r}^{normal}$, $f_{r}^{a_{i}}$, and n are the deviation in the resonating frequency, the resonating frequency at a normal incidence angle, the resonating frequency at various incidence angles, and the number of incidence angles, respectively. Figure 8a,b show the various incident angle (0° to 85°) curves in the proposed structure’s TE and TM modes. For each incident signal (0° to 85°) from the source, the proposed structure reflects the signal in the same frequency with minimum frequency shifting, as shown in Figure 8a,b. With this collected data, we can identify the frequency deviation. The calculated angle mean deviation (AMD) depicts the minimum percentage of frequency deviation 0.45% for TE mode and 1.01% for TM mode.

Metasurfaces are an increasingly important technology for manipulating electromagnetic waves. One crucial aspect of metasurface design is polarization insensitivity. This means that the polarization of the incoming wave does not affect the device’s performance. Polarization sensitivity is a common issue with traditional devices that can reduce efficiency and performance in real-world scenarios where the polarization of the wave is unknown or uncontrolled. Polarization insensitivity enables the design of devices and systems that can operate with a wide range of incident polarizations, providing greater flexibility and adaptability. To achieve polarization insensitivity, the metasurface must have a stable operating frequency for both polarized incident waves. Figure 9 shows the S_21_ curve for the meta-atom with different polarizations, following a normal incidence angle of 65°, considered for angular stability. At various angles, the unit cell has a stable operating frequency.

The proposed system improves polarization insensitivity, with an average maximum deviation rate of approximately 0.027% for different polarized orientations. This demonstrates the system’s stability and performance, making it a more versatile and efficient platform for manipulating electromagnetic waves.

Based on the above analysis, it is evident that the proposed structure satisfies both angular and polarization stability, making it effective for reconfigurable applications.

#### 3.1.2. Investigation of Reconfigurable Metasurface

Recently, there has been a growing interest in improving signal coverage by developing hardware solutions. One such solution is a reconfigurable metasurface, which consists of low–cost passive elements that can be programmed electronically to manipulate the phase of incoming electromagnetic waves [33]. This technology offers a promising approach to controlling and shaping the direction of radio waves. The PIN diodes can be used to implement this scheme with the proposed structure. Instead of using a PIN diode within the unit cell, the diode can be placed between two elements. Four PIN diodes can be added to a 2 × 2 metasurface to achieve phase reconfigurability. Using PIN diodes ensures that the resonating frequency remains the same while the phase changes. With the help of the four diodes, it is possible to achieve 2^4^ combinations of phase shifting.

The PIN diode can be analyzed through its equivalent circuits in both forward- and reverse-bias states, as shown in Figure 10. A reconfigurable metasurface analysis is performed to investigate the phase changes in the metasurface. The diodes’ simple biasing circuit is applied with a voltage of >2 V. It has been observed that when the PIN diodes are in different bias states, there are two configurations of a single switch of the metasurface. This can be explained as follows: when the diode is forward-biased (RON = 5.2 Ω and LON = 0.01 nH), two meta-atoms of the 2 × 2 metasurface become interconnected due to the slight resistance of the diode and undergo some phase changes. Conversely, when the diode is reverse-biased (C_OFF_ = 0.03 pF and L_OFF_ = 0.01 nH), the small capacitance lowers the total capacitance of each element, the elements cannot be interconnected, and other phases are changed.

Figure 11 shows that the frequency band remains constant during the reverse- and forward-bias of all four diodes in the 2 × 2 metasurface, providing a reflection band between 11 and 20 GHz. By biasing these four diodes, 16 different phase changes can be achieved. This can be visualized in Figure 12, and the details of the phase changes can be found in Table 2.

Table 2 reflects the different states of the reconfigurable metasurface. The state where the diode is in reverse-bias (OFF state) is represented by ‘0’, whereas forward-bias (ON state) is represented by ‘1’. Combining D1, D2, D3, and D4 allows you to achieve different phase changes for the resonating frequency of 15 GHz. The maximum phase difference the proposed metasurface can attain is 10°, and the minimum is 4°. You can also apply the same switching to a 32 × 32 array of meta-elements to achieve phase reconfigurability. For this, you can use four diodes for each 2 × 2 array of the 32 × 32 metasurface through the conductive vias of each element. You can provide different biasing for all the diodes. For instance, the activation of the first switch in every 2 × 2 array of the 32 × 32 metasurface can be achieved by the forward biasing of D1. Depending on the incident angle of the signal, the element is activated and reflects the pattern shown in Figure 13.

### 3.2. Second Layer

The second layer in the RIS is the copper plane. Its purpose is to block the transmitted wave and effectively reflect the signal. When electromagnetic waves interact with the ground plane, they can create standing waves, diffraction effects, and other interference patterns that interfere with the metasurface’s intended behavior. However, if designed properly, a copper plane can achieve specific effects in combination with the metasurface. For example, it can be used for beam steering or reflecting waves in a desired direction. The copper plane should be added at some distance from the reflecting surface to achieve this. The distance between the reflecting surface and the conducting plane should be a fraction of the wavelength of the operating frequency. The simulation setup Figure 14 shows an analysis of the distance between the reflecting metasurface and the copper plane.

The setup depicted in Figure 14 includes two horn antennas: one for transmission (antenna 1) and one for reception (antenna 2). By directing the incident signal onto the reflecting surface with the copper plane, the signal is completely reflected, thus allowing for control of the transmission. Analysis of the distance (S) between the metasurface and copper plane shown in Figure 15 revealed that complete reflection was achieved at a distance of 18 mm.

### 3.3. Third Layer

The third layer of the RIS is equipped with an intelligent control module and communication interface. The control module regulates the phase shifts of the passive reflecting elements. It employs software algorithms to determine the ideal configurations of the reflecting elements based on real–time channel conditions and system requirements. This intelligent control module enables a dynamic reconfiguration of the reflective elements, allowing for the adaptive optimization of signal propagation in real time. The communication interface acts as the link between the RIS and the rest of the wireless communication system. It facilitates seamless integration and operation by enabling the exchange of control signals and data between the RIS and other network components. The main focus is controlling the phase shifting elements for the passive reflecting elements, which requires an Arduino, keypad, and voltage controller.

The algorithm of the process involves receiving input from the keypad, selecting the state according to the key pressed, and providing the appropriate biasing to the switching circuit. With the switching state, the phase shift can be attained. Table 3 explains the detailed switching relation with the keypad and PIN diodes, and the connection of the circuit is shown in Figure 16.

The keypad buttons 0 to F can control 16 different biasing conditions in the table. When the keypad is used, the PIN diodes adjust to their respective state and change phase. The architecture of an RIS consists of passive reflecting elements, a copper plane, and a control module, which work together to manipulate electromagnetic signals and optimize communication performance in 6G systems.

## 4. Experimental Validation and Evaluation

To validate the performance of the proposed work, a prototype of a 32 × 32 reflecting metasurface has been fabricated for experimental evaluation, as shown in Figure 17a. The metasurface was created using printed circuit board (PCB) technology on Rogers RT/Duroid 5880 substrates, with the same parameters as the structure. In addition, the reflecting metasurface was patterned on the top surface of the substrate. For the reconfigurable metasurface, PIN diode MADP–000907–4020P, shown in Figure 17b, was used, with four diodes used for each 2 × 2 array element. Figure 17c shows the microcontroller used to control the PIN diodes.

During the measurement setup, the proposed architecture of the RIS was positioned with absorbers surrounding it. The Monostatic Radar Cross Section (RCS) arrangements, which are affixed on a holder, consist of ridged horn antennas capable of functioning at frequencies ranging from 12 to 18 GHz. The identical antennas utilized in measurement setup 1 were employed in this experiment. The device under test was positioned in front of the horn, as depicted in Figure 18. Experiments were conducted in the anechoic chamber to assess the metasurface’s reflection properties and minimize external interference and antenna reflections during the measurements. The physical dimensions of the prototype are 14.4 cm, with an array size of 14.4 cm × 14.4 cm.

Figure 19 compares the measured and simulated results of a movable handheld microwave from Agilent Technologies (N9951A) and an EM simulation tool. The results confirm the efficient functioning of the proposed structure in reflecting the signal. In Figure 20a,b, the angular stability of the reflecting metasurface is shown for the angle range of 0° to 85° for both TE and TM modes of polarization, respectively. The results demonstrate a high level of agreement for the incident angles with an impressively low AMD% of 0.48% and 1.08% for TE and TM modes. The maximum deviation between simulation and measurement is about 3%, which may result from fabrication tolerance. Despite some frequency deviations, the measurement results show a remarkable agreement with the simulations, thus validating the results.

The proposed structure is biased by the microcontroller through a level shifter (1.4 V) using wires that are welded on top of the metallic layers. When the keypad is used, the diodes in the metasurface become forward–biased and cause phase changes. Figure 21 illustrates the phase graph of the proposed structure in a few states. The measured results were compared with the simulation results and showed a variation of only 1% from the simulation. By studying the graph, it is possible to see that the RIS works efficiently with phase reconfigurability of a maximum of 10° and a minimum of 4°.

The phase changes for each state were measured and are recorded in Table 4. The three-layer RIS architecture was successfully designed and developed, with simulation results matching those measured.

Table 5 compares existing works regarding the performance parameters with the proposed work. To the author’s knowledge, a single-layer high angularly stable-phase reconfigurable reflective metasurface is not available for RIS architecture. It was observed that the proposed active design has numerous advantages over different reported works in the literature in terms of angular stability, coding type, number of layers, and polarization insensitivity. Another issue with the metasurface designs is the angular stability. The proposed design has 85° angular stability compared to the other active designs.

Moreover, many structures and multiple layers complicate the design, affecting the system’s reliability and performance. The proposed work has an effective single–layer structure. Even though [19] contains a single-layer metasurface, the switching characteristics have only 2-bit coding. The achieved range of angular stability and re-configurability evidence the suitability of the proposed work as a base IRS.

## 5. Conclusions

The proposed work showcases the successful development and testing of a three-layer architecture for a reconfigurable reflecting metasurface at a frequency of 15 GHz. This innovative design utilizes a square loop, octagonal loop, and tripole metasurface to achieve high customization, reduced coupling, and precise phase steering. Furthermore, integrating diodes and a digital phase controller demonstrates its effectiveness in attaining precise phase control and polarization adjustment, resulting in improved signal quality and reduced degradation. This simple single–layered breakthrough design is a significant milestone in electrical engineering and has immense potential to contribute to future technological advancements in IRS physical layer design.

## Figures and Tables

**Figure 1 sensors-23-09166-f001:**
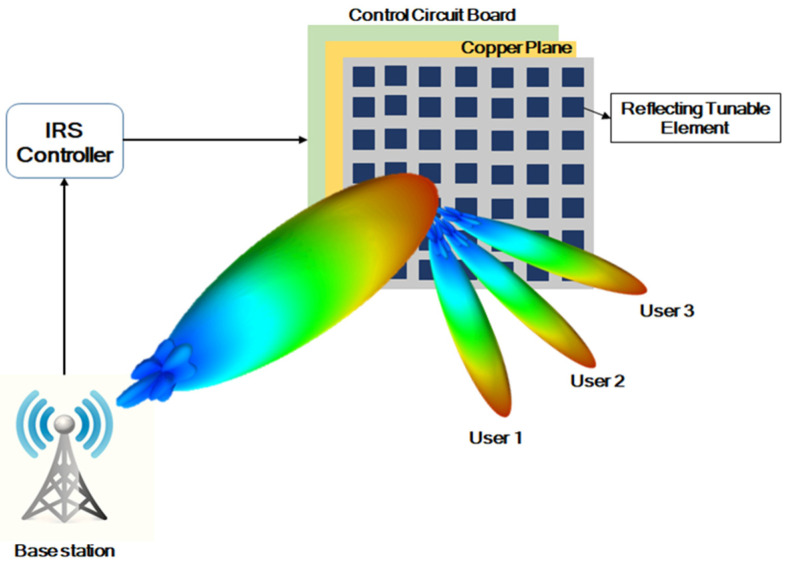
Architecture of RIS.

**Figure 2 sensors-23-09166-f002:**
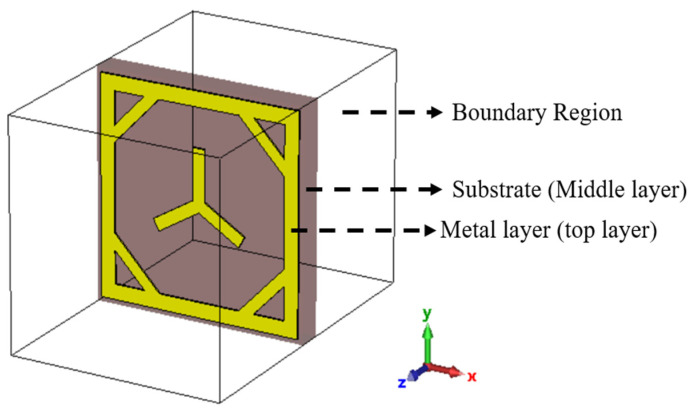
Structure of unit meta-atom.

**Figure 3 sensors-23-09166-f003:**
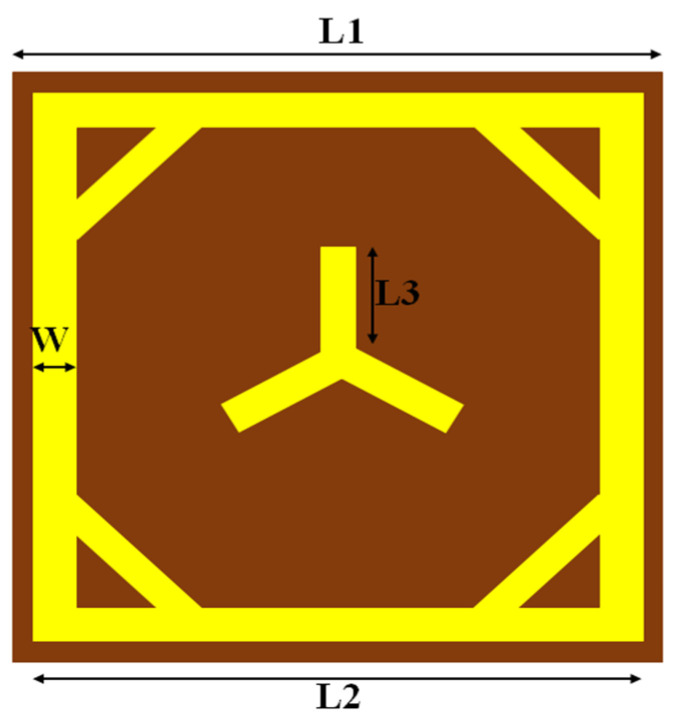
Combine loop meta-atom with tripole.

**Figure 4 sensors-23-09166-f004:**
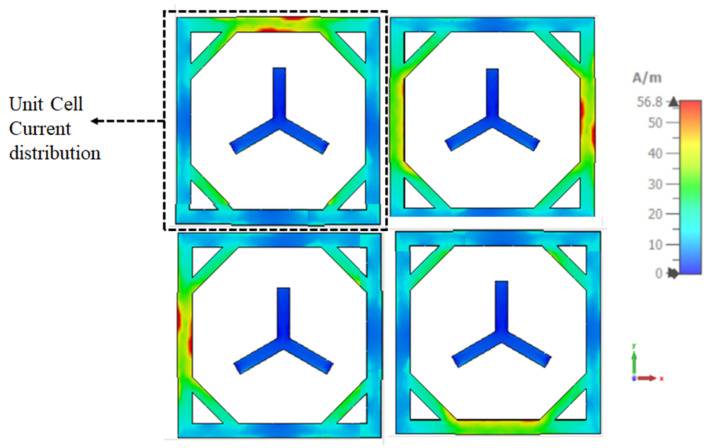
Current distribution of 2 × 2 metasurface.

**Figure 5 sensors-23-09166-f005:**
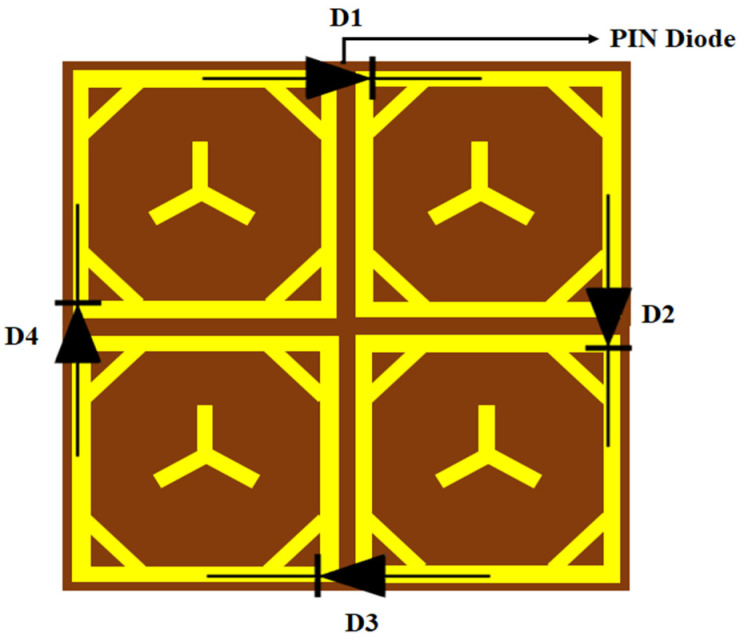
Reconfigurable of 2 × 2 metasurface with PIN diode equivalent circuit.

**Figure 6 sensors-23-09166-f006:**
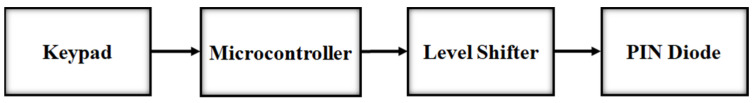
Block diagram of switch controller.

**Figure 7 sensors-23-09166-f007:**
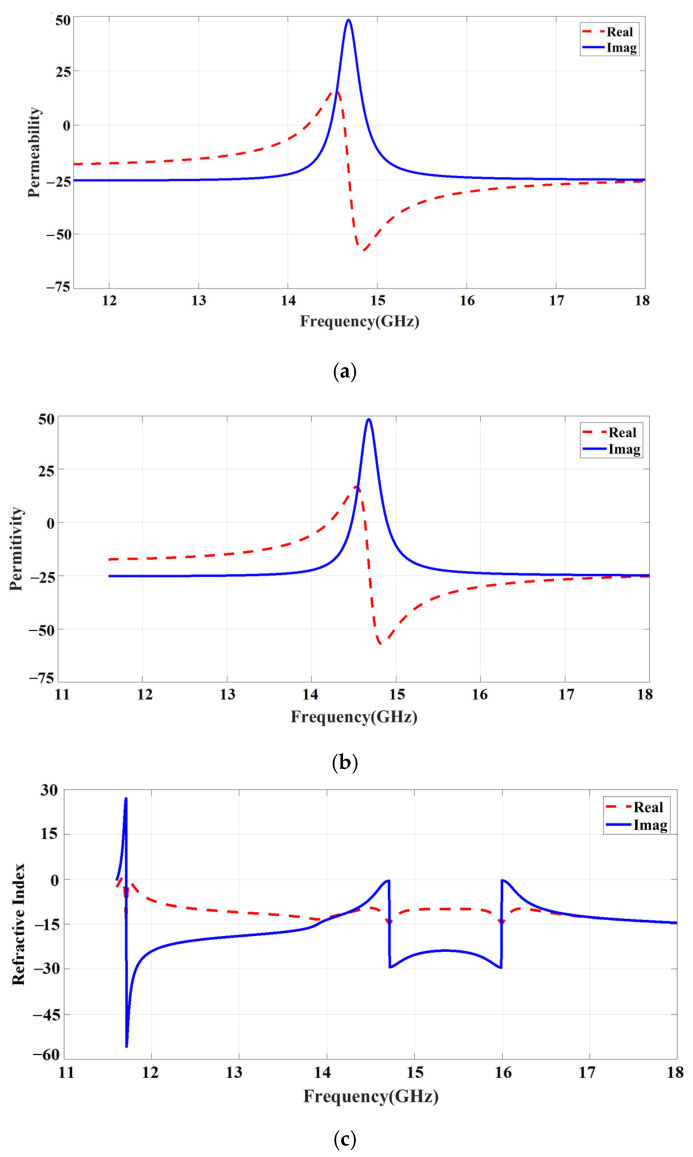
Metasurface analysis with effective medium: (**a**) effective permeability; (**b**) effective permittivity; (**c**) effective refractive index.

**Figure 8 sensors-23-09166-f008:**
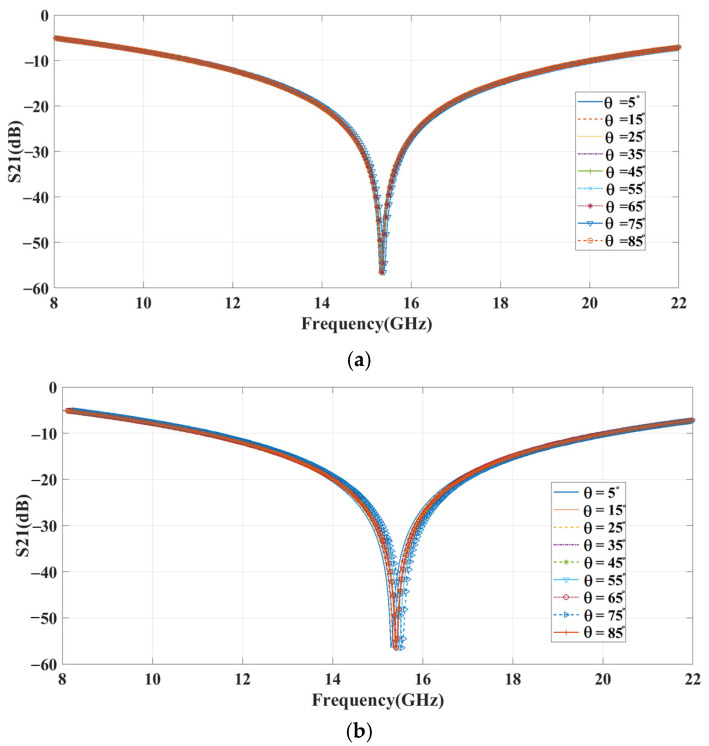
Incident angle analysis: (**a**) TE mode; (**b**) TM mode.

**Figure 9 sensors-23-09166-f009:**
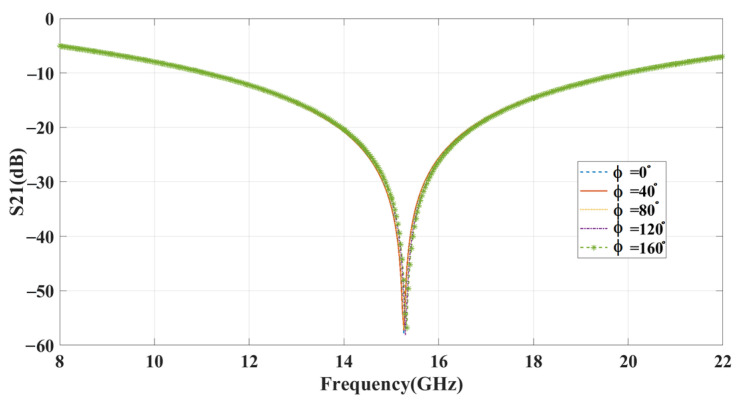
Polarization analysis for the incident angle of 65°.

**Figure 10 sensors-23-09166-f010:**
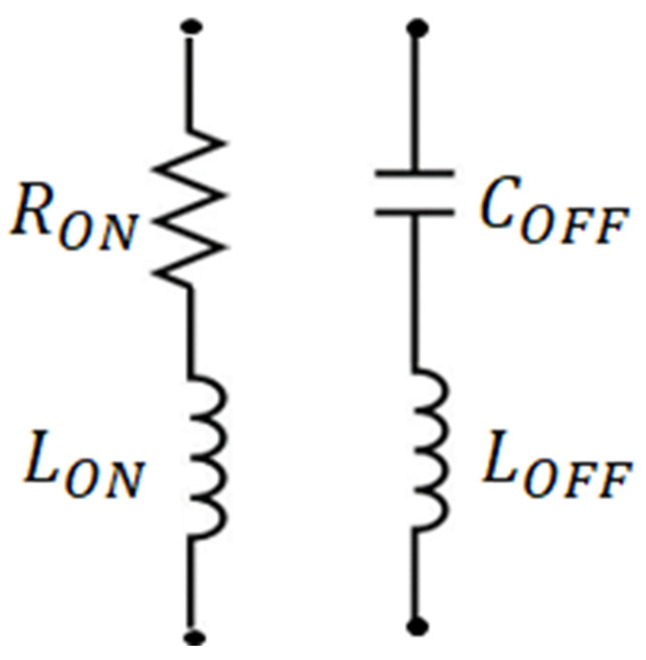
Equivalent circuit for PIN diode.

**Figure 11 sensors-23-09166-f011:**
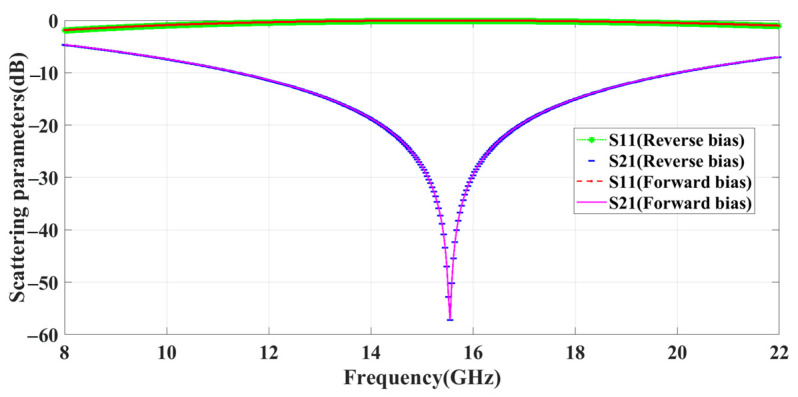
Scattering parameter results using with and without bias.

**Figure 12 sensors-23-09166-f012:**
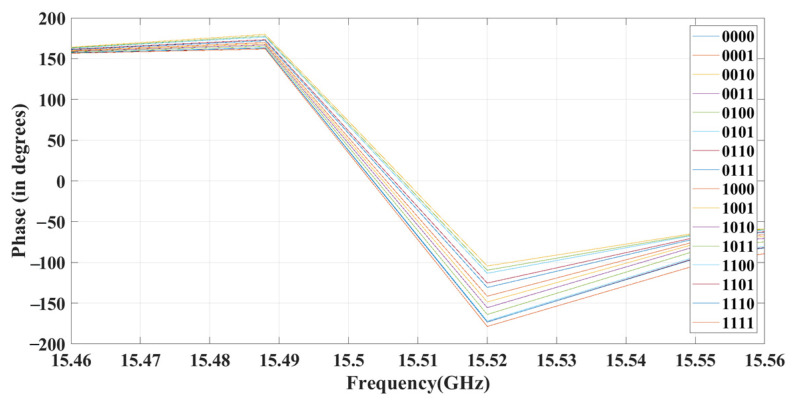
Phase changing results for various switching states.

**Figure 13 sensors-23-09166-f013:**
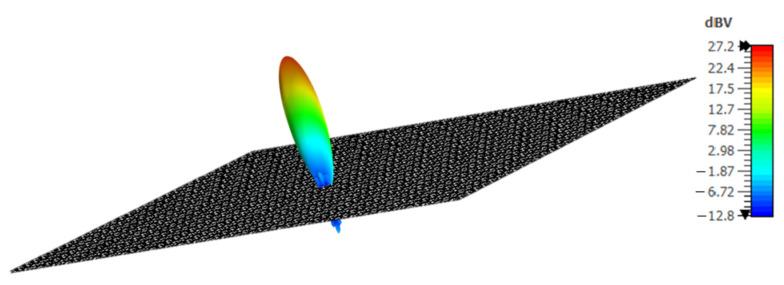
Pattern for 32×32 metasurface.

**Figure 14 sensors-23-09166-f014:**
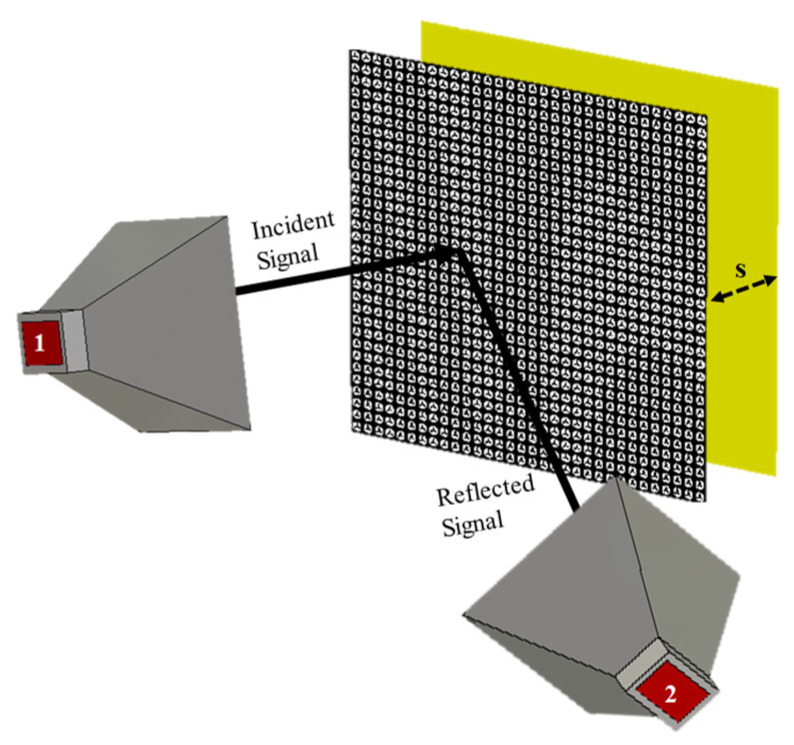
Metasurface simulation setup with horn antennas.

**Figure 15 sensors-23-09166-f015:**
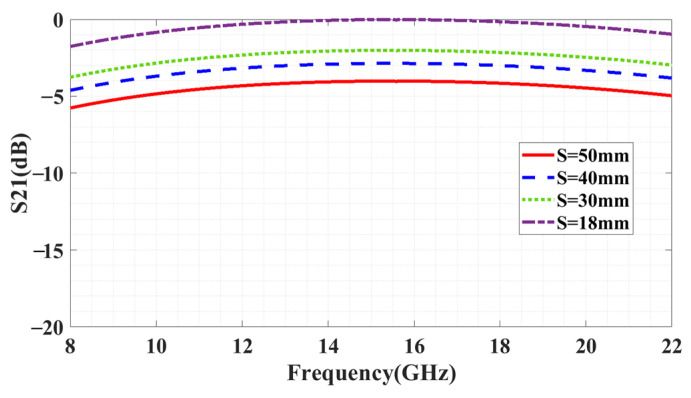
Distance analysis between surface and copper plane.

**Figure 16 sensors-23-09166-f016:**
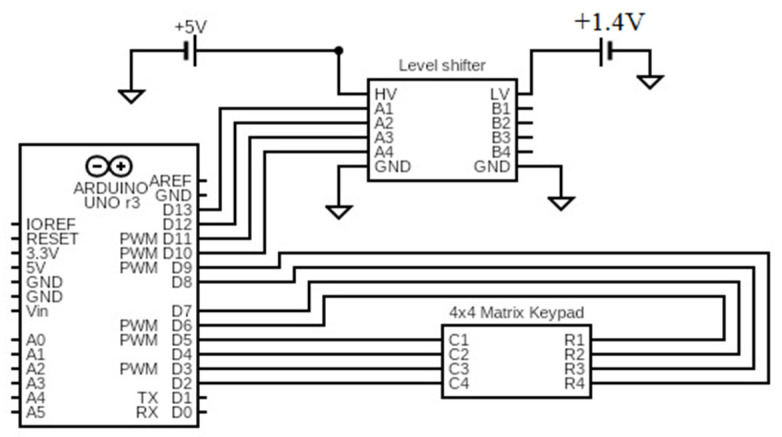
The architecture of microcontroller for controlling PIN diodes.

**Figure 17 sensors-23-09166-f017:**
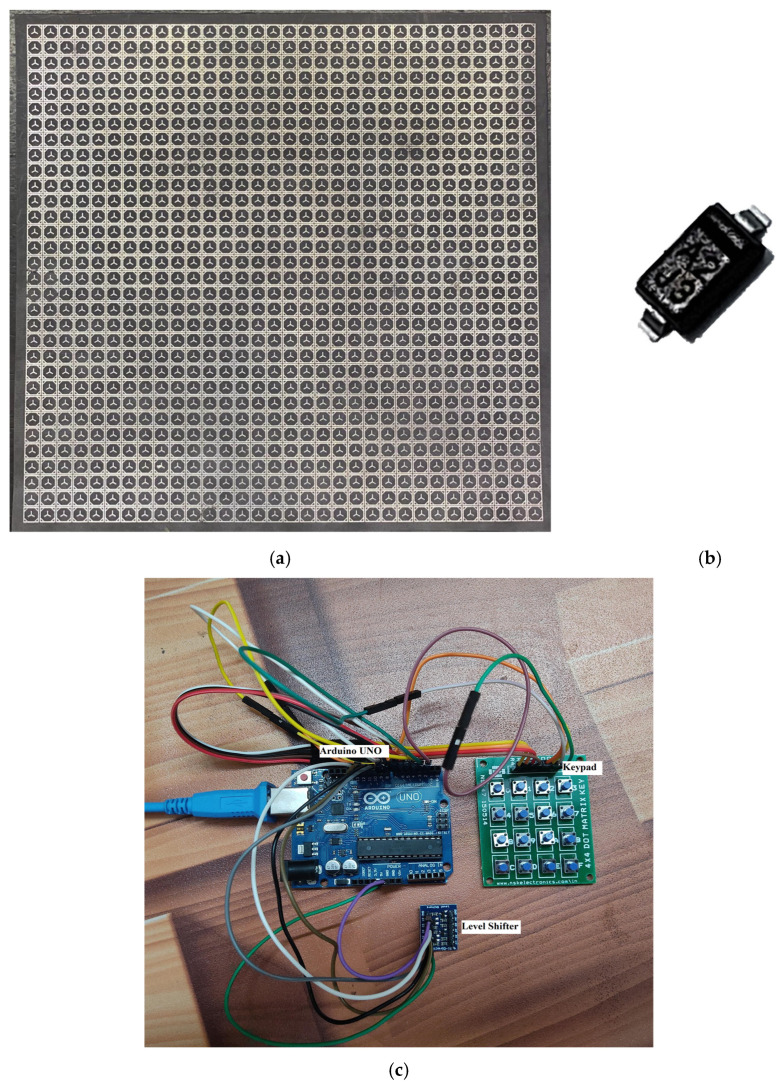
Prototype of the proposed structure. (**a**) Metasurface, (**b**) PIN diode, (**c**) switch controller.

**Figure 18 sensors-23-09166-f018:**
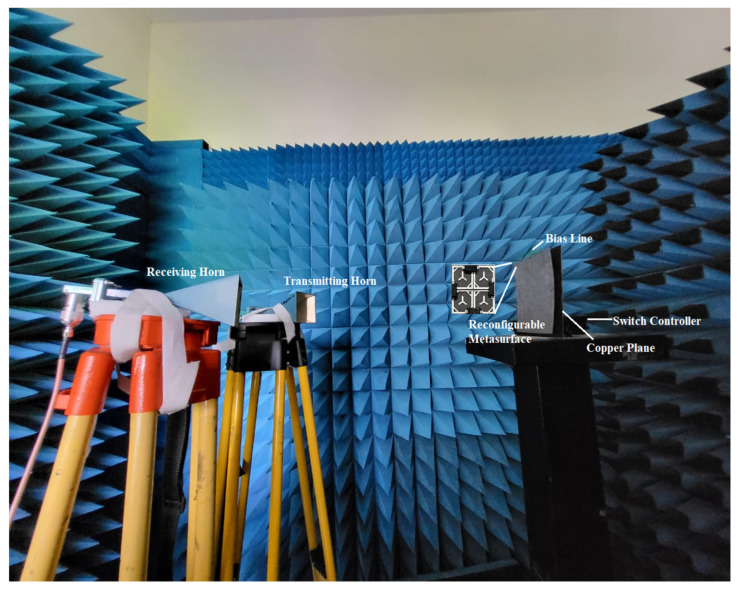
Measurement setup.

**Figure 19 sensors-23-09166-f019:**
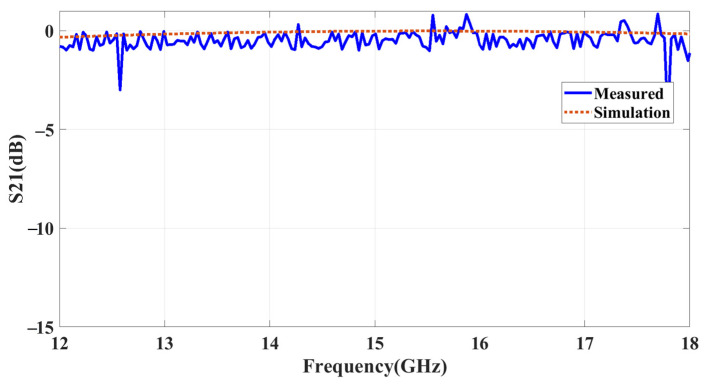
Measured and simulated results of reflecting surface.

**Figure 20 sensors-23-09166-f020:**
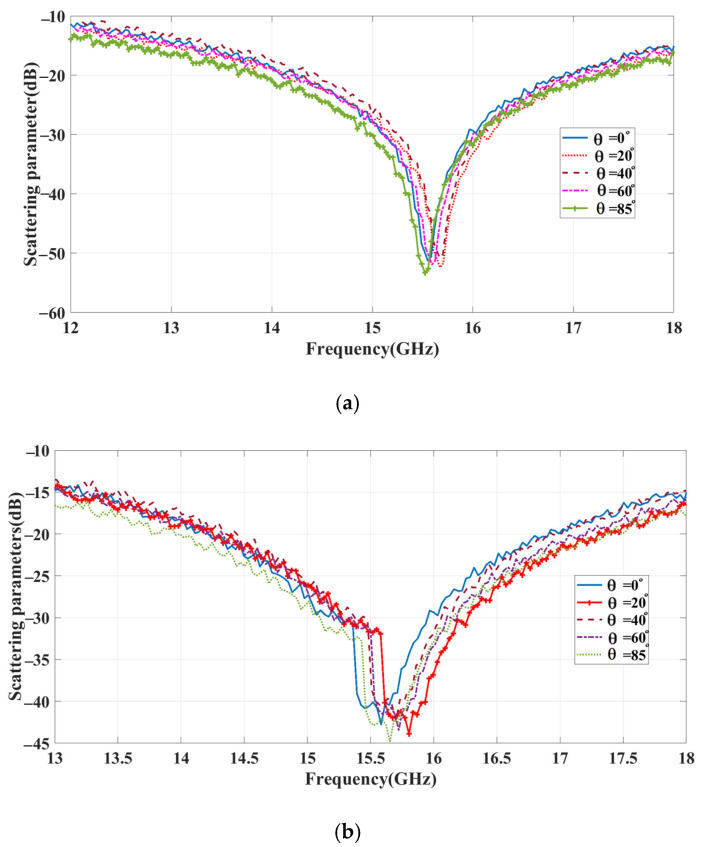
Measured angular analysis: (**a**) TE mode; (**b**) TM mode.

**Figure 21 sensors-23-09166-f021:**
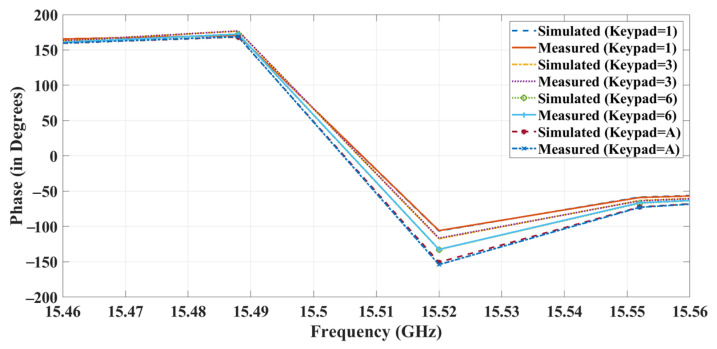
Comparison phase results of proposed work.

**Table 1 sensors-23-09166-t001:** Parameters of meta-atom.

**Parameter**	L1	L2	W	L3	t
**Dimensions (mm)**	4.5	4.3	0.3	1.075	0.508

**Table 2 sensors-23-09166-t002:** Phase reconfigurable metasurface results.

State				Frequency (GHz)	Phase Change (In Degrees)
D1	D2	D3	D4		
0	0	0	0	15.48	−102
0	0	0	1	15.53	−106
0	0	1	0	15.53	−111
0	0	1	1	15.48	−117
0	1	0	0	14.47	−123
0	1	0	1	15.48	−132
0	1	1	0	15.47	−136
0	1	1	1	15.48	−140
1	0	0	0	15.48	−145
1	0	0	1	15.53	−153
1	0	1	0	15.47	−157
1	0	1	1	15.48	−166
1	1	0	0	15.47	−170
1	1	0	1	15.48	−180
1	1	1	0	15.48	−189
1	1	1	1	15.53	−196

**Table 3 sensors-23-09166-t003:** Relation between keypad and PIN diodes.

Keypad	States				Phase Change (In Degrees)
	D1	D2	D3	D4	
0	OFF	OFF	OFF	OFF	−102
1	OFF	OFF	OFF	ON	−106
2	OFF	OFF	ON	OFF	−111
3	OFF	OFF	ON	ON	−117
4	OFF	ON	OFF	OFF	−123
5	OFF	ON	OFF	ON	−132
6	OFF	ON	ON	OFF	−136
7	OFF	ON	ON	ON	−140
8	ON	OFF	OFF	OFF	−145
9	ON	OFF	OFF	ON	−153
A	ON	OFF	ON	OFF	−157
B	ON	OFF	ON	ON	−166
C	ON	ON	OFF	OFF	−170
D	ON	ON	OFF	ON	−180
E	ON	ON	ON	OFF	−189
F	ON	ON	ON	ON	−196

**Table 4 sensors-23-09166-t004:** Comparison of simulation results with experimental results.

Keypad	Simulation Results	Experimental Results
0	−102	−103
1	−106	−105
2	−111	−111
3	−117	−116
4	−123	−125
5	−132	−132
6	−136	−135
7	−140	−140
8	−145	−144
9	−153	−150
A	−157	−156
B	−166	−165
C	−170	−172
D	−180	−181
E	−189	−189
F	−196	−200

**Table 5 sensors-23-09166-t005:** Comparison of proposed work with recent related works.

Ref.	Unit Cell Type	Layer	Control Mechanism	Angular Stability	Polarization Insensitivity	Coding Type
[18]	Patch	Multi-layer	PIN diode	60°	N/A	2-bit
[19]	Patch	Single layer	PIN diode	N/A	N/A	2-bit
[21]	2-bit	Multi-layer	Two PIN diode	60°	N/A	2-bit
[22]	Patch with parasitic patch	Multi-layer	PIN diode	60°	N/A	1-bit
[23]	2-bit	Multi-layer	PIN diode	35°	N/A	2-bit
[24]	Patch	Multi-layer	PIN diode	50°	N/A	1-bit
[25]	Cross-dipole-shaped	Multi-layer	PIN diode	N/A	N/A	2-bit
This work	Combined loop with tripole	Single Layer	PIN Diode	85°	Yes	4-bit

## Data Availability

Data are contained within the article.

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
