# Peer review of "Reconfigurable Metasurface: Enabling Tunable Reflection in 6G Wireless Communications"

_sensors, 2023, doi:10.3390/s23229166_

Round 1

Reviewer 1 Report

Comments and Suggestions for Authors

This work describes a reconfigurable combined loop metasurface for RIS. This technology can potentially improve communication coverage and quality in 6G applications. The metasurface can be reconfigured to achieve various phase reflections by controlling the state of PIN diodes. A 32x32 metasurface has been validated through numerical simulations and experiments, demonstrating its potential for use in 6G applications. Several flaws in the work must be addressed before it is considered for publication.

The paper has several typos. Authors need to proofread the paper to eliminate all of them.

The word 'our' should not be used in a research paper as it is too informal.  

The pronoun 'we' is used too many times in the paper. I'd say that generally, 'we' is appropriate to discuss future work in the conclusion, but besides that, it should be used sparingly.  

Some sentences are too long. Generally, writing short sentences with one idea per sentence is better.

The literature review is incomplete. Several relevant references still need to be added.

The introduction should clearly explain the fundamental limitations of prior work relevant to this paper.

This reviewer could not grasp what the contributions of this study are. It seems like the work has no original contributions, being just a reproduction of works in the literature.  Contributions should be highlighted more. It should be made clear what is novel and how it addresses the limitations of prior work.

There is no related work section. This section will help the readers and reviewers understand how this world stands out from the body of literature.

The authors should add a table that compares the key characteristics of prior work to highlight their differences and limitations. The authors may also consider adding a line in the table to describe the proposed solution.

A reconfigurable metasurface solution is presented, but it is essential to explain better the design decisions (e.g., why the solution is designed like that).

Comments on the Quality of English Language

The paper has several typos. Authors need to proofread the paper to eliminate all of them.

The word 'our' should not be used in a research paper as it is too informal.  

The pronoun 'we' is used too many times in the paper. I'd say that generally, 'we' is appropriate to discuss future work in the conclusion, but besides that, it should be used sparingly.  

Some sentences are too long. Generally, writing short sentences with one idea per sentence is better.

Author Response

Response to Reviewer 1 Comments

1. Summary

Thank you for your time in reviewing the manuscript. Please find the responses and revisions below, with corresponding highlights in the re-submitted files.

2. Questions for General Evaluation

Reviewer’s Evaluation

Response and Revisions

Does the introduction provide sufficient background and include all relevant references?

Yes/Can be improved/Must be improved/Not applicable

We have considered the reviewer's feedback and added more relevant references to the manuscript.

Are all the cited references relevant to the research?

Yes/Can be improved/Must be improved/Not applicable

Thank you for your concern. We have included a separate section for the literature review and discussed recent similar contributions and our contribution's scope.

Is the research design appropriate?

Yes/Can be improved/Must be improved/Not applicable

Thank you! To the author's knowledge, we have presented all the necessary design resources.

Are the methods adequately described?

Yes/Can be improved/Must be improved/Not applicable

We have added a detailed methodology to provide better clarity on our contribution.

Are the results clearly presented?

Yes/Can be improved/Must be improved/Not applicable

By accounting for the reviewer’s comment, we compared our contribution with similar research works for better clarity.

Are the conclusions supported by the results?

Yes/Can be improved/Must be  improved/Not applicable

We improvised the conclusion with respect to our research findings.

3. Point-by-point response to Comments and Suggestions for Authors

Comments 1: The literature review is incomplete. Several relevant references still need to be added.

Response 1: We express our gratitude for your valuable suggestion, and we would like to inform you that we have duly considered it. As per your recommendation, we have included the relevant references in Section 1.1 detailed the recent and relevant references. This change can be found in page number 2 (Introduction Section) & 3 (Literature Survey) in the revised manuscript.

Comments 2: The introduction should clearly explain the fundamental limitations of prior work relevant to this paper.

Response 2: Thanks for the Reviewers comments. As the Reviewer commented, we have included an additional literature review section explaining the fundamental limitations of prior work relevant to this paper and it is reflected in page number 3.

Comments 3: This reviewer could not grasp what the contributions of this study are. It seems like the work has no original contributions, being just a reproduction of works in the literature.  Contributions should be highlighted more. It should be made clear what is novel and how it addresses the limitations of prior work.

Response 3: We acknowledge the reviewer's comment on the choice of the proposed design. Although we employed traditional structures, the choice of structures and their integration methodology are the authors’ contribution to achieving the desired results. The methodology is explained on page 4. Also, the key contribution is included in the last paragraph of the Introduction section.

Comments 4: There is no related work section. This section will help the readers and reviewers understand how this world stands out from the body of literature.

Response 4: We express our gratitude for your valuable suggestion, and we would like to inform you that we have duly considered it. As per your recommendation, we have included a separate literature review section (1.1) and discussed recent similar contributions.

Comments 5: The authors should add a table that compares the key characteristics of prior work to highlight their differences and limitations. The authors may also consider adding a line in the table to describe the proposed solution.

Response 5: Thank you for your valuable suggestion. We have incorporated the comparison table by comparing the key characteristics of the proposed solution with existing works. We updated the manuscript by including table 5 on page no. 18.

Comments 6: A reconfigurable metasurface solution is presented, but it is essential to explain better the design decisions (e.g., why the solution is designed like that).

Response 6: We acknowledge the reviewer's comment regarding the choice of proposed design decisions. After conducting a thorough literature survey, we have come to understand that the RIS system requires a design that ensures low power consumption, simplicity, and angular stability while being insensitive to polarization. Our research has been focused on designing a simple structure that achieves better angular stability and polarization insensitivity. We have updated the manuscript and explained the significance of our contribution on page 4.

4. Response to Comments on the Quality of English Language

Point 1: The paper has several typos. Authors need to proofread the paper to eliminate all of them

Response 1: Thank you for your feedback. We have taken great care to correct any typos in the revised manuscript.

Point 2: The word 'our' should not be used in a research paper as it is too informal.  

Response 2: Thank you for your guidance. I would like to inform you that we have made the necessary corrections to the statements in the revised manuscript. 

Point 3: The word 'our' should not be used in a research paper as it is too informal.  

Response 3: Thank you for your guidance and I would like to inform you that we have made the necessary corrections to the statements in the revised manuscript. 

Point 4: Some sentences are too long. Generally, writing short sentences with one idea per sentence is better.

Response 3:Thank you for your feedback. We have taken great care to present each idea in a simple, short sentence.

Reviewer 2 Report

Comments and Suggestions for Authors

Author Response

1. Summary

Thank you for your time in reviewing the manuscript. Please find the responses and revisions below, with corresponding highlights in the re-submitted files.

2. Questions for General Evaluation

Reviewer’s Evaluation

Response and Revisions

Does the introduction provide sufficient background and include all relevant references?

Yes/Can be improved/Must be improved/Not applicable

We have considered the reviewer's feedback and added more relevant references to the manuscript.

Are all the cited references relevant to the research?

Yes/Can be improved/Must be improved/Not applicable

Thank you for your concern. We have included a separate section for the literature review and discussed recent similar contributions and our contribution's scope.

Is the research design appropriate?

Yes/Can be improved/Must be improved/Not applicable

Thank you!

Are the methods adequately described?

Yes/Can be improved/Must be improved/Not applicable

We have added a detailed methodology to provide better clarity on our contribution.

Are the results clearly presented?

Yes/Can be improved/Must be improved/Not applicable

The feedback provided by the reviewer has been carefully considered and incorporated into the revised interpretation of the results.

Are the conclusions supported by the results?

Yes/Can be improved/Must be  improved/Not applicable

We improvised the conclusion with respect to our research findings.

3. Point-by-point response to Comments and Suggestions for Authors

Comments 1: It is suggested to briefly introduce the challenges at the beginning of the abstract to make readers more clear.

Response 1: We express our gratitude for your valuable suggestion, and we would like to inform you that we have duly considered it and summarized the challenges in the abstract, page 1.

Comments 2: Both the motivations and contributions should be further improved. The motivation is missing. The authors need to clarify why they investigated this work and the advantages and novelty of their proposed scheme compared to existing ones.

Response 2: Thank you for your suggestion. We have considered the reviewer's feedback and made the necessary changes to the paper. Specifically, we added a new literature review section on page 3, paragraph 3, which explains the limitations of prior work relevant to our research. We also highlighted our contribution on page 2. Thank you again for your valuable input.

Comments 3: It is suggested to introduce the following recent works in RIS [R1]-[R4] fields to highlight the state-of-art of this paper:

[R1] “Pain without gain: Destructive beamforming from a malicious RIS perspective in IoT networks,” IEEE Internet of Things Journal, early access, Sep. 2023, DOI: 10.1109/JIOT.2023.3316830.

[R2] “Refracting RIS aided hybrid satellite-terrestrial relay networks: Joint beamforming design and optimization,” IEEE Transactions on Aerospace and Electronic Systems, vol. 58, no. 4, pp. 3717-3724, Aug. 2022.

[R3] “Active RIS assisted rate-splitting multiple access network: Spectral and energy efficiency tradeoff,”

IEEE Journal on Selected Areas in Communications, vol. 41, no. 5, pp. 1452-1467, May 2023.

[R4] “Secure satellite transmission with active reconfigurable intelligent surface,” IEEE Communications Letters, vol. 26, no. 12, pp. 3029-3033, Dec. 2022.

Response 3:. As the Reviewer commented, the recent state-of-art and related references are included and appropriately cited in the revised manuscript.

Comments 4: The authors should explain why they adopt the diodes placed between the two elements of the metasurface shown in Figure 5? And how could they propose this design?

Response 4: We express our gratitude for your valuable suggestion. Based on the high current flow in the metasurface, the position of diodes in the proposed work was determined. Figure 4 shows the positions where the current flow is high, and diodes were placed at those locations to achieve a symmetric distribution. By controlling the diode, we can control the current distribution, which aids in phase reconfiguration. This explanation is reflected while describing the Fig 4 & Fig 5.

Comments 5: The authors should compare the performance of the proposed RIS antenna with existing one in simulations to show its advantages.

Response 5: Thank you for your valuable suggestion. As suggested, we have incorporated the comparison table (Table 5) by comparing the key characteristics of the proposed solution with existing works in page 18.

Reviewer 3 Report

Comments and Suggestions for Authors

The topic of the manuscript is an intelligent reflective surface (IRS) to enhance the throughput and coverage of 5G and 6G wireless communication networks. Specific solutions for a phase reconfigurable metasurface with (as one might assume) a novel combination of PIN diodes are presented.
The article is interesting and in a good form as, after a knowledge analysis, the proposed solution is described, which was developed using simulation tools, the results of which are presented in the content. Finally, and importantly, the performance was confirmed with tests on a real object.

Comments:
In my opinion there is a lack of a clear description indicating what specifically the proposed solution stands out from existing solutions, the reader can only guess.

It seems to me that some descriptions are redundant. Specifically, the introductions to the individual chapters in Section 3 seem to duplicate information described in Section 2.

In the case of Figure 8, I miss a description for the less initiated reader of what is implied by these graphs.

Why the operating frequency is 15.4 GHz. Was this frequency deliberately chosen and if so why. Or did it just come out that way in the proposed structure.

Figure 11 does not show the green and blue lines, so why the conclusions described below.

What I find missing from the conclusions is a justification as to why this is a breakthrough solution.

Author Response

Response to Reviewer 3 Comments

1. Summary

Thank you for your time in reviewing the manuscript. Please find the responses and revisions below, with corresponding highlights in the re-submitted files.

2. Questions for General Evaluation

Reviewer’s Evaluation

Response and Revisions

Does the introduction provide sufficient background and include all relevant references?

Yes/Can be improved/Must be improved/Not applicable

We have considered the reviewer's feedback and added more relevant references to the manuscript.

Are all the cited references relevant to the research?

Yes/Can be improved/Must be improved/Not applicable

Thank you! 

Is the research design appropriate?

Yes/Can be improved/Must be improved/Not applicable

Thank you! To the author's knowledge, we have presented all the necessary design resources.

Are the methods adequately described?

Yes/Can be improved/Must be improved/Not applicable

Thank You!

Are the results clearly presented?

Yes/Can be improved/Must be improved/Not applicable

The feedback provided by the reviewer has been carefully considered and incorporated into the revised interpretation of the results.

Are the conclusions supported by the results?

Yes/Can be improved/Must be  improved/Not applicable

We improvised the conclusion with respect to our research findings.

3. Point-by-point response to Comments and Suggestions for Authors

Comments 1: In my opinion there is a lack of a clear description indicating what specifically the proposed solution stands out from existing solutions, the reader can only guess.

Response 1: Thank you for your valuable suggestion. As the Reviewer commented, the main contribution is included in Introduction section of the revised manuscript. Further, we have incorporated the comparison table (Table 5 in page 5) by comparing the key characteristics of the proposed solution with existing works.

Comments 2: It seems to me that some descriptions are redundant. Specifically, the introductions to the individual chapters in Section 3 seem to duplicate the information described in Section 2.

Response 2: Thank you for your suggestion. We have considered the reviewer's feedback and made the necessary changes to the revised paper under sections 2 and 3.

Comments 3: In the case of Figure 8, I miss a description for the less initiated reader of what is implied by these graphs.

Response 3:. Thanks! We have addressed the reviewer's suggestion by incorporating a detailed description of Figure 8 on page 9 of the updated manuscript.

Comments 4: Why the operating frequency is 15.4 GHz. Was this frequency deliberately chosen and if so why. Or did it just come out that way in the proposed structure.

Response 4: The study of literature provides valuable insights. Meanwhile, there is an ongoing discussion about allocating operating bands for 6G. Although it has not been finalized yet, a session presented in National Telecommunication Institute for Policy Research, Innovation & Training (NTIPRIT) and Qualcomm's proposal suggest that X-band (8-12 GHz) and Ku-band (12-18 GHz) are being considered for supporting wide area use cases. Therefore, we are focusing on addressing the Ku-band with an operating frequency of 15.4 GHz in our proposed work.

Comments 5: Figure 11 does not show the green and blue lines, so why the conclusions described below.

Response 5: Thanks for bringing this to our attention. We utilized different line representations for better clarity in visualizing all the projected legends in Figure 8.

Comments 6: What I find missing from the conclusions is a justification as to why this is a breakthrough solution.

Response 6: We concur with the reviewer's feedback. To address this point, we have revised the conclusion section of our paper to reflect the implications of our research findings more accurately. We express our gratitude for your valuable comment.

Round 2

Reviewer 2 Report

Comments and Suggestions for Authors

The authors have well addressed all my concerns, no further comments.